# Availability of labile carbon controls the temperature-dependent response of soil organic matter decomposition in alpine soils

Dario Püntener[1], Tatjana C. Speckert[1], Yves-Alain Brügger[1], and Guido L.B. Wiesenberg[1]

[1]Department of Geography, University of Zurich, Winterthurerstrasse 190, CH-8057 Zurich, Switzerland

**Correspondence:** Dario Püntener (dario.puentener@geo.uzh.ch)

**Abstract.** Soil organic matter (SOM) decomposition in alpine environments is influenced by multiple factors including temperature and substrate quality. It is important to understand how these factors influence soil carbon dynamics. We incubated subalpine forest and pasture soils at 12.5 °C, 16.5 °C, and 20.5 °C for one year with and without addition of fresh grass litter to assess impacts on total organic carbon (TOC) and lignin dynamics. In the absence of litter, TOC losses were limited, accounting for $6.7 \pm 2.4\%$ in forest soils and $3.3 \pm 1.6\%$ in pasture soils after 360 days, with no consistent temperature effect. In contrast, litter addition strongly increased the decomposition of primary SOM, resulting in TOC losses of $11.8 \pm 1.1\%$ in forest soils and $17.4 \pm 1.9\%$ in pasture soils, which were higher at elevated temperatures. Lignin concentrations declined markedly in forest soils, indicating that warming increases the decomposition of more complex SOM. Pasture soils were dominated by the breakdown of more labile litter-derived C. These results demonstrate that substrate availability is a stronger control for SOM decomposition than temperature. Increasing litter inputs in combination with rising temperatures could accelerate SOM decomposition, potentially shifting subalpine soils from carbon sinks to sources under future climate scenarios, irrespective of vegetation cover.

## 1 Introduction

Soils represent one of the largest terrestrial reservoirs of organic carbon (C) containing an estimated 2000 to 2700 Pg C (Batjes, 2016; Jackson et al., 2017). Soil systems are pivotal components of the global carbon cycle, acting both as significant sources and sinks of carbon dioxide ($CO_2$) by storing large amounts of organic carbon or releasing it through decomposition processes (Schmidt et al., 2011). The decomposition of soil organic matter (SOM) is an important factor that controls carbon fluxes to the atmosphere, directly impacting atmospheric $CO_2$ levels and influencing climate change (Conant et al., 2011; Crowther et al., 2016). This balance between carbon storage and release is particularly sensitive to temperature, with recent studies showing that rising temperatures can shift soils from being net carbon sinks to significant carbon sources (Crowther et al., 2016). Rising temperatures are expected to accelerate microbial activity, leading to faster SOM decomposition and increased carbon release from soils (Davidson and Janssens, 2006; Chen et al., 2024). This process not only reduces the capacity of soils to act as carbon sinks but also amplifies climate change by creating a positive feedback loop (Davidson and Janssens, 2006; Conant et al., 2011). Long-term warming experiments indicate that the decomposition of labile carbon pools occurs rapidly under elevated temperatures, and sustained warming can destabilize more complex and stable carbon pools, which were previously

thought to be more resistant to microbial breakdown (Melillo et al., 2017; Hicks Pries et al., 2017; Ofiti et al., 2023; Zosso et al., 2023). This suggests that soil carbon stocks are highly vulnerable to warming (Bright et al., 2025), with significant implications for global carbon budgets.

Alpine regions, which store substantial stocks of soil organic carbon (SOC), are especially vulnerable to warming (Bonfanti et al., 2025). These high-altitude ecosystems are characterized by unique conditions, including low temperatures, short growing seasons, and slow SOM decomposition rates, which historically have promoted the accumulation of organic carbon in soils (Zierl and Bugmann, 2007; Hiltbrunner et al., 2013). These regions are characterized by a suite of vegetation types ranging from lower-elevation forests to shrublands, grasslands, and pastures (Grabherr et al., 2010). In comparison to other ecosystems, alpine soils are more sensitive to climatic changes (Hock et al., 2019) due to their reliance on seasonal snow cover, limited vegetation inputs, and the predominance of organic matter with chemical structures that require specific microbial or enzymatic pathways for decomposition (Marschner et al., 2008; Djukic et al., 2010; Schmidt et al., 2011), e.g. the complex polymer lignin (Bahri et al., 2008). These characteristics not only make alpine soils significant carbon reservoirs, but also increase their vulnerability to warming-induced carbon losses.

Climate change is occurring more rapidly in alpine regions than in many other parts of the world, leading to significant environmental transformations (Beniston, 2003; Rogora et al., 2018). Rising temperatures (Hock et al., 2019), declining snow cover (Klein et al., 2016), extended growing periods (Rogora et al., 2018), and upward shifts in the tree line (Gehrig-Fasel et al., 2007) are altering vegetation composition and ecosystem dynamics (Hagedorn et al., 2019). Forest encroachment into alpine grasslands drives changes in SOM inputs and soil properties, affecting both the build-up and decomposition of organic matter (Hagedorn et al., 2019). As warming extends the short growing season during which microbial activity occurs, the decomposition of SOM accelerates, leading to greater $CO_2$ emissions from alpine soils (Hiltbrunner et al., 2013). Labile SOM fractions decompose rapidly under warmer conditions, while the decomposition of more chemically complex SOM components, such as lignin, may depend on shifts in microbial community composition and the availability of labile carbon substrates to fuel enzymatic activity (Fissore et al., 2013; Walker et al., 2018).

In this study, we investigate how increased temperatures influence the breakdown of fresh litter inputs and native SOM in alpine forest and pasture soils. Using a one-year laboratory incubation experiment under controlled conditions, we simulate projected climate warming scenarios to assess temperature-driven changes in SOM dynamics for soils derived from alpine grassland and coniferous forest sites. By examining the interplay between soils that developed under different vegetation types and temperature, this research aims to provide critical insights into the vulnerability of alpine soils to warming and their broader implications for global carbon cycling and climate change mitigation strategies. The primary research questions and hypotheses are:

1. How does decomposition differ in alpine soils developed under pasture and coniferous forest vegetation when exposed to increasing temperature?

2. What is the influence of litter input on SOM decomposition in forest and pasture soils under varying temperature?

3. How do interactions between litter input and temperature affect the stability and decomposition of organic carbon in alpine soils developed under pasture and coniferous forests?

Increased decomposition of SOM with rising temperatures is expected for both alpine forest and pasture soils (Nottingham et al., 2020; Soong et al., 2021), with a more pronounced stimulation in the pasture soil due to higher microbial activity and greater availability of easily decomposable SOM compared to the forest soil (Dirnböck et al., 2003; Hiltbrunner et al., 2013; Canedoli et al., 2020). In contrast, the coniferous forest soil, characterized by a fungal-dominated decomposition and the presence of more complex SOM such as lignin, will show a slower but steady increase in decomposition with warming (Ortiz et al., 2016), leading to gradual yet persistent carbon release over time.

Litter input in the form of grass will enhance SOM decomposition by providing fresh, easily decomposable SOC that stimulates microbial activity (Kuzyakov et al., 2000), especially in these subalpine soils (Guo et al., 2022). In the forest soil, this may promote the enhanced breakdown of older, more complex SOM such as lignin (Ibanez et al., 2021), whereas the alpine pasture soil may show a weaker increase in decomposition due to reduced C-limitation (Li et al., 2018).

The interaction between litter input and temperature will further reduce SOC stability in these alpine soils, as increasing temperatures enhance SOM mineralisation (Prietzel et al., 2016). This effect is expected to be strongest in the forest soil, where the combination of warming and litter input could destabilize older carbon pools with more complex SOM such as lignin, leading to sustained C loss over time (Tian et al., 2016; Blanco et al., 2023).

## 2 Materials and methods

### 2.1 Study site and sample preparation

Soil material was collected on a south-facing slope above the village of Jaun, Canton of Fribourg, Switzerland [7°15'54 E; 46°37'17 N] from a pasture and an adjacent forest site. The two sites are located at altitudes between 1500 and 1550 m a.s.l. Mean air temperature reaches from 0.6 °C in winter to 12.5 °C in summer with mean annual precipitation of 1250 mm with maximal precipitation in summer (Hiltbrunner et al., 2013). According to the World Reference for Soil Resources (WRB) (IUSS Working Group WRB, 2015), soils were classified as Leptic Eutric Cambisol Clayic on a calcareous bedrock with clay-dominated texture for both the pasture site (60 % clay, 30 % silt, 8 % sand) as well as the forest site (50 % clay, 35 % silt, 12 % sand (Speckert et al., 2023)). The soils are acidic with a pH only slightly differing between the two sites with pH 5.08 for the pasture and pH 4.83 for the forest soil. The pasture site has been grazed by cattle during the summer months (May-September) (Hiltbrunner et al., 2013). The plant community consists mainly of herbaceous species with dominant occurrences of ribgrass (*Plantago lanceolata* L.) and reed fescue (*Festuca arundinacea* Schreb.). The forest site is dominated by Norway spruce (*Picea abies* L.) with tree ages of at least 130 years (Speckert et al., 2023). Mineral soil samples were collected July 2020 on an area of ca. 1 m$^2$ at a depth of 5–10 cm after removal of organic layers (forests) and surface mineral soil with high root frequencies. Overall, one composite sample (ca. 30 kg) of soil was collected for each of both sites. All replicate 50 g incubation subsamples

derive from these two composites. Soils were sieved for <2 mm and visible root remains were removed by tweezers. Thereafter, soils were homogenized by manual mixing with a hand shovel.

## 2.2 Incubation setup

To investigate differences in organic matter decomposition between soils that developed under different vegetation cover, forest and pasture soil samples were incubated in closed jars in temperature-controlled incubators (Panasonic MIR-554-PE). The influence of elevated temperature on the decomposition of organic material was targeted and therefore, soils were incubated under three different temperatures. The lowest temperature of 12.5 °C ($T_{12.5}$), acting as the control temperature, corresponds to the 2015–2020 average air temperature of the growing season between May and September of the sampling site (weather station Jaun-Forchen by WSL-SLF (2021)). This temperature was chosen as low temperature in this experiment and not the mean annual air temperature as the predominant decomposition of SOM is taking place during the warmer summer season and slows down during the winter season (Yao et al., 2011; Žifčáková et al., 2016), where the temperature at the sampling site is close to freezing (Hiltbrunner et al., 2013). The two treatments with increased temperatures of 16.5 °C (+4 °C, $T_{16.5}$) and 20.5 °C (+8 °C, $T_{20.5}$) correspond to the expected temperature increases in the European Alps predicted for the years 2080–2099 with emission scenario RCP8.5 (Hock et al., 2019). The incubation follows mainly the approach described in Abiven and Andreoli (2011). For each temperature treatment, 48 samples (24 forest soil, 24 pasture soil), each weighing 50 g, were placed in 2-l glass jars. At the beginning, 20 ml of water were added, corresponding to the field capacity of the soils, and vials containing 20 ml water were placed in the jars aside the soil to ensure increase of humidity in the air space of the jars and avoid drying of the soil (see setup in Supplement Fig. S1). A pre-incubation of the samples for 18 days was conducted to stabilize and test the activity of the microbial community. After pre-incubation, 1.25 g of dried ca. 1-2 cm long cut leaf tissues from perennial ryegrass (*Lolium perenne*) grown in a $^{13}$C enriched atmosphere (Studer et al., 2017) was added to 17 forest samples and 17 pasture samples for each temperature treatment. The carbon added to the soil by the litter addition was approximately equal to the carbon already present in the soil samples, resulting in a fresh to old carbon ratio of 1:1. The incubation ran for the period of 360 days. At different times throughout the incubation, a subset of the incubated soil samples was destructively sampled. An overview of the sampling scheme can be found in the Supplement Fig. S2. During destructive sampling, samples were placed in plastic bags and immediately transferred to a freezer (-28°C). The incubated soil samples were freeze-dried to a constant weight and milled in a horizontal ball mill (MM400, Retsch, Germany).

## 2.3 Carbon and nitrogen analysis

To asses total carbon and nitrogen concentrations (TC, TN) of each sample, as well as stable carbon isotope composition ($\delta^{13}$C), 5 mg of the soil material were weighed into tin capsules and measured using an elemental analyzer coupled to an isotope ratio mass spectrometer (EA-IRMS; Flash 2000-HT Plus, linked by Conflo IV to Delta V Plus isotope ratio mass spectrometer, Thermo Fisher Scientific, Germany). The concentrations of carbon and nitrogen and the stable isotope composition were calibrated using a soil reference material (Haplic Chernozem, Harsum, Germany; University of Zurich 2023) as laboratory

internal standards and IAEA-600 caffeine as certified standard. At least two analytical replicates were measured for each sample.

## 2.4 Lignin analysis

The soil material was subjected to alkaline CuO oxidation procedure by Hedges and Ertel (1982) to break down the lignin polymer into its different monomers. An adapted version of the microwave digestion by Goñi and Montgomery (2000) was used (Heim and Schmidt, 2007). Approximately 600 mg of soil material was oxidized with 500 mg of CuO powder, 50 mg of ammonium iron-(II)-sulfate and 20 ml 2M NaOH in $N_2$ flushed microwave tubes at 150 °C for 90 minutes and subsequent cooling down. To each sample, an internal standard of 500 $\mu$l of cinnamic acid and ethylvanillin mix (each with a concentration

of 1 g l⁻¹) was added. Solids were removed by centrifuging for 4 minutes at 3000 rpm and following decanting. The supernatant was adjusted to pH 2.10 with 32% HCl. The samples were subsequently collected on preconditioned (ethyl acetate, methanol, water) DSC-18 SPE columns and eluted with 5 x 500 $\mu$l ethyl acetate. Residual water was removed with $Na_2SO_4$ and the samples were dried under $N_2$ and then redissolved with 400 $\mu$l of internal standard solution (1 g l⁻¹ anisic acid in ethyl acetate). Quantification of individual lignin monomers was performed after derivatization of 70 $\mu$l sample with 70 $\mu$l BSTFA/TCMS

99:1 derivatization reagent for 20 minutes at 60 °C. The analysis was performed by gas chromatography-flame ionization detection (GC-FID; 7890B GC System, Agilent, USA). A DB-5MS column (Agilent, USA; length 50 m, internal diameter 200 $\mu$m, film thickness 0.33 $\mu$m) was used with the following temperature program: Start at 80 °C, hold for 5 min, ramp to 110 °C with +2 °C min⁻¹, ramp to 170 °C with +0.5 °C min⁻¹, ramp to 320 °C with +15 °C min⁻¹, hold 10 min. Injection was done using a multimode inlet running in splitless mode (temperature program: start at 90 °C, hold for 0.5 min, ramp to 400 °C with

+850 °C min⁻¹, hold 2 min). Compound identification was done by measurement under the same chromatographic conditions as explained above but analysed by gas chromatography mass spectrometry (GC-MS, 6890N GC System, 5973N MS System, Agilent, USA and by comparison to known standards and Wiley/NIST spectral libraries. Losses due to sample preparation were corrected using the cinnamic acid and vanillic acid internal standards (Heim and Schmidt, 2007).

To measure compound-specific stable carbon ($\delta^{13}$C) isotope composition, samples were analyzed in triplicate using a gas

chromatograph (TRACE 1310, Thermo Scientific, Germany) coupled to a Delta V Plus isotope ratio mass spectrometer via GC-Isolink II and ConFlo IV (Thermo Fisher Scientific, Germany). The shift in the isotopic composition introduced by adding trimethylsilyl carbon during derivatization was corrected using the mass balance equation by Dignac et al. (2005) (Equation (1)):

$$\delta_{UD} = \frac{n_D}{n_{UD}}\delta_D - \frac{n_{BSTFA}}{n_{UD}}\delta_{BSTFA} \tag{1}$$

where $\delta_{UD}$ represents the isotopic ratio of the underivatized phenol, $n_D$ is the number of C atoms in the derivatized phenol, $n_{UD}$ the number of C atoms in the underivatized phenol, $\delta_D$ is the isotopic ratio of the derivatized phenol (measured on GC-IRMS), $n_{BSTFA}$ is the number of C atoms added from BSTFA (depending on the phenol) and $\delta_{BSTFA}$ is the isotopic ratio of BSTFA (measured with GC-IRMS and compared to underivatized standards).

Natural abundance isotope ratios in samples without litter addition are expressed as $\delta$ $^{13}$C relative to the international Vienna

Pee Dee Belemnite (VPDB). In labelled samples with litter addition, the enrichment is expressed in units of atom % excess (APE):

$$APE = (atom\%)_{L+} - (atom\%)_{L-} \tag{2}$$

with $(atom\%)_{L+}$ as the concentration of $^{13}C$ of the labelled samples and $(atom\%)_{L-}$ as the concentration of $^{13}C$ of the samples without litter addition (Slater et al., 2001). For $(atom\%)_{L-}$, two different averaged values were used for forest and

pasture soil, respectively.

## 2.5 Statistics

All statistical analyses were performed using R software version 4.4.1 (R Core Team, 2024). Prior to analysis, all data were tested for normality and homogeneity of variances using the Shapiro-Wilk test and Levene's test, respectively. Where necessary, data were log-transformed to meet assumptions of parametric tests.

To assess the effects of litter addition (with vs. without), vegetation type (forest vs. pasture), temperature (control vs. elevated), and incubation time (short-term vs. long-term) on SOM decomposition, a four-way analysis of variance (ANOVA) was conducted. This model allowed us to test for both main effects and interaction effects between factors. When significant interaction terms were detected, post-hoc pairwise comparisons were conducted using Tukey's Honest Significant Difference (HSD) test to further explore specific differences between treatment combinations.

To evaluate the temporal dynamics of SOM decomposition, repeated measurements of ANOVA were applied to examine changes over the different incubation time points, considering litter presence, vegetation type, and temperature as between-subject factors.

All statistical tests were two-tailed, with a significance level set at $\alpha = 0.05$. Results are presented as means $\pm$ standard error of the mean (SEM), unless otherwise indicated.

# 3 Results

## 3.1 Total organic carbon concentrations

At the beginning of the incubation experiment, total organic carbon (TOC) concentrations of forest soil samples without litter addition (L$^-$) averaged at $43.8 \pm 1.6$ mg g$^{-1}$ across all temperature treatments (Fig. 1(a)). Initial TOC was slightly higher for pasture L$^-$ soils ($45.5 \pm 1.1$ mg g$^{-1}$). We observed a decrease for forest L$^-$ soils during the incubation period of 360 days (-6.7

$\pm 2.4$ %, $p = 0.012$), which was almost twice as high as for the pasture L$^-$ soils (-3.3 $\pm 1.6$ %, $p = 0.053$). Until the end of the incubation experiment, there was no difference in the temporal trend between different temperature treatments for forest and pasture L$^-$ soils ($p = 0.39$; $p = 0.52$). Litter addition (L$^+$) initially increased TOC of forest (+21.5 $\pm 1.2$ %, $p < 0.001$) and pasture (+26.9 $\pm 2.0$ %, $p < 0.001$) soils (Fig. 1(b)). The average decrease of TOC of forest L$^+$ and pasture L$^+$ soils was significantly stronger in comparison with the respective L$^-$ soils. Forest L$^+$ soils showed an almost twofold decrease compared to forest L$^-$

soils (-11.8 $\pm 1.1$ %) during the incubation period. For the L$^+$ pasture soils, the decrease was almost sixfold compared to the

pasture $L^-$ soils (-17.4 $\pm$ 1.9 %). Generally, we observed for both, forest and pasture $L^+$ soils, decreasing TOC with increasing temperature. Differences between different temperature treatments, however, were only significant at certain points during the incubation period (see *p*-values in Supplement Table S4).

### 3.2 Total organic carbon isotope composition

Forest $L^-$ soils ($\delta^{13}C$ -25.8 $\pm$ 0.02 ‰) were slightly less depleted in $^{13}C$ (Fig. 1(g)) compared to pasture $L^-$ soils ($\delta^{13}C$ -26.6 $\pm$ 0.01 ‰, $p < 0.001$). No change in $\delta^{13}C$ was detected in forest $L^-$ soils during the incubation period. For pasture $L^-$ soils, we observed a slight decrease of $\delta^{13}C$ ($p = 0.01$), mainly due to slightly increased values for $T_{12.5}$ after 14 and 28 days. No temperature effect was visible in $L^-$ soils for both, forest and pasture soil samples (see Supplement Table S5). Litter addition (Fig. 1(h)) increased initial $^{13}C$ for both soils to a similar degree (forest: +0.56 $\pm$ 0.03 %, $p < 0.001$; pasture: +0.58 $\pm$ 0.03 %, $p < 0.001$). In $L^+$ soils, significant decreases in atomic % excess (APE) $^{13}C$ were observed during the incubation period for all treatments (see Supplement Table S1). A much stronger decrease was noted during the first 28 days of the experiment, which was less pronounced for the remainder of the incubation period (Supplement Table S2).

The influence of temperature on the decomposition rates varied between forest and pasture. Forest soil samples exhibited a relatively constant ratio between short (28 d) - and long-term (360 d) decomposition rates across different temperatures. In contrast, in the pasture soil we could detect a slight increase in this ratio with temperature.

$L^+$ soils showed a clear trend with temperature, with both pasture and forest having a stronger decrease of APE$^{13}C$ at higher temperatures ($p < 0.001$). This trend was visible at most time points during the incubation, especially between $T_{12.5}$ and the increased temperatures (see Supplement Table S5).

### 3.3 Nitrogen concentrations

Initial total nitrogen (TN, Fig. 1(c)) concentrations were on average lower in the forest $L^-$ soil than in the pasture $L^-$ soil (3.4 $\pm$ 0.1 mg g$^{-1}$, 4.7 $\pm$ 0.1 mg g$^{-1}$, respectively). TN did not change in $L^-$ soils during the incubation. Litter addition increased initial TN for forest and pasture soil, which was more pronounced for forest soil than for pasture soil samples (+15.6 $\pm$ 1.8 %, $p < 0.001$; +10.7 $\pm$ 1.9 %, $p < 0.001$). In $L^+$ soils (Fig. 1(d)), an increase was observable from day 0 to day 56 for the $T_{12.5}$ treatment, from day 0–28 for $T_{16.5}$ treatment and for day 0–14 for $T_{20.5}$ treatment. Thereafter, TN dropped until the end of the experiment or stayed almost constant after they first dropped ($T_{20.5}$ pasture). We saw mostly differences between $T_{16.5}$ and $T_{20.5}$ in the beginning of the incubation and at a later stage between $T_{12.5}$ and the elevated temperature treatments (see Supplement Table S6).

### 3.4 Carbon to nitrogen ratios

We observed significantly higher initial carbon to nitrogen ratios (C/N, Fig. 1(e)) in forest $L^-$ soils compared to pasture $L^-$ soils (12.8 $\pm$ 0.1, 9.6 $\pm$ 0.05, $p < 0.001$). During the whole incubation experiment, this difference remained similar. The C/N ratio decreased significantly during the incubation period for forest $L^-$ (-5.3 $\pm$ 1.0 %, $p = 0.002$) and pasture $L^-$ soils (-3.6 $\pm$ 0.6 %,

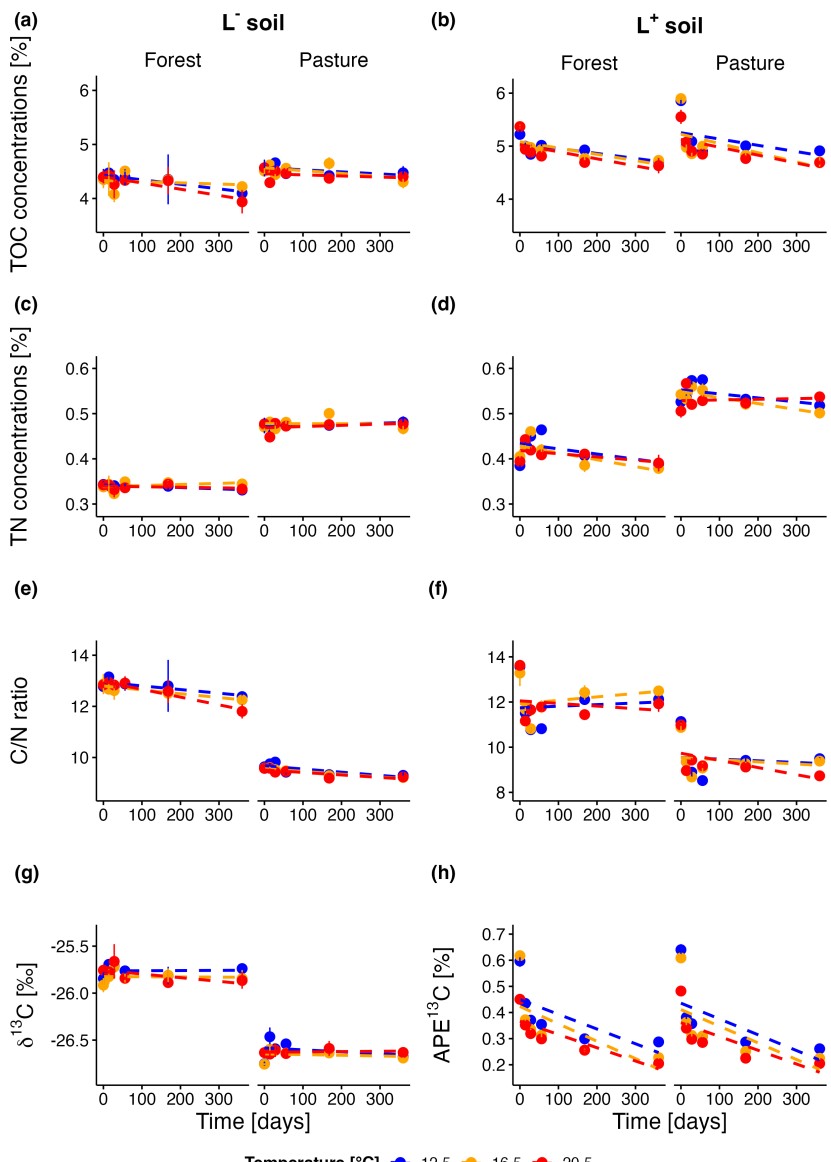

**Figure 1.** TOC concentrations over the incubation period for forest and pasture soils without (L⁻; (a)) and with litter addition (L⁺; (b)); TN concentrations over the incubation period for forest and pasture soils without (L⁻; (c)) and with litter addition (L⁺; (d)); C/N ratio evolution over the incubation period for forest and pasture soils without (L⁻; (e)) and with litter addition ((L⁺; (f)); Natural abundance $\delta$ $^{13}$C over the incubation period for forest and pasture soils without litter addition ((L⁻; (g)); Atomic % excess of $\delta$ $^{13}$C over the incubation period for forest and pasture soils with litter addition. (L⁺; (h))

$p < 0.001$). There were only differences between different temperatures (Supplement Table S7) after 28 days in pasture L⁻ soils between $T_{12.5}$ and $T_{16.5}$ ($p = 0.002$), and $T_{12.5}$ and $T_{20.5}$ ($p = 0.006$). Litter addition (Fig. 1(f)) led to an initial increase of the

C/N ratio for both forest and pasture L$^+$ soils, with a more pronounced increase for the latter (+5.1 ± 1.4 %, $p = 0.01$; +14.6 ±
0.8 %, $p < 0.001$). During the incubation period, the C/N ratio decrease was more pronounced in L$^+$ than L$^-$ soils, with forest
L$^+$ soils showing a lower decrease than pasture L$^+$ soils (-9.7 ± 1.5 %, $p < 0.001$; -16.3 ± 0.0 %, $p < 0.001$). While the decrease
of the C/N ratio in L$^-$ soils was almost consistent over the entire incubation period, it showed a different pattern in L$^+$ soils. We
observed a strong decrease during the first 28 days followed by a much less pronounced decrease or even slight increase of the
C/N ratio until the end of the incubation period. Throughout most of the incubation period in both forest and pasture soils, a
significant temperature trend was observed, with higher temperatures associated with lower C/N ratios (Supplement Table S7).

### 3.5 Phenol concentrations

Total phenol concentrations averaged over all temperatures were higher in forest L$^-$ compared to pasture L$^-$ soils with initial
concentrations of 1509.5 ± 95.1 $\mu$g g$^{-1}$ and 1260.9 ± 63.4 $\mu$g g$^{-1}$ (Fig. 2). In L$^+$ soil, we observed higher phenol concentrations
for both forest and pasture soils (1760.4 ± 56.0 $\mu$g g$^{-1}$, 1465.4 ± 43.4 $\mu$g g$^{-1}$, respectively, Fig. 3). During the incubation period,
phenol concentrations decreased significantly in all treatments. We observed a stronger decrease for forest L$^+$ and L$^-$ soils (-
31.1 ± 1.5 %, $p = 0.009$; -21.7 ± 1.7 %, $p = 0.097$) compared to pasture L$^+$ and L$^-$ soils (-24.1 ± 1.2 %, $p < 0.001$; -15.0 ±
0.9 %, $p = 0.25$). To test the influence of temperature on phenol concentrations, we compared the concentration decrease with
different temperatures of different phenol groups: Vanillyl (vanillin, vanillic acid, acetovanillone), syringyl (syringaldehyde,
syringic acid, acetosyringone), cinnamyl (coumaric acid, ferulic acid), and p-hydroxyl (salicylaldehyde, salicylic acid, piceol)
phenols. For the individual groups, we could detect significant decreases over time for vanillyl (forest L$^-$, $p = 0.040$), syringil
(forest L$^-$, $p = 0.016$; pasture L$^+$, $p = 0.005$), cinnamyl (all treatments, $p < 0.001$) and p-hydroxyl (forest L$^-$, $p = 0.006$; forest
L$^+$, $p = 0.009$; pasture L$^-$, $p = 0.045$) phenol concentrations. We could not detect significant differences between different
temperature treatments for total phenol concentrations. For the individual phenol groups, only cinnamyl phenols in pasture L$^+$
soils showed a significant temperature dependency ($p = 0.005$) with differences between T$_{12.5}$ and T$_{16.5}$, and T$_{12.5}$ and T$_{20.5}$ ($p$
= 0.010, $p = 0.011$).

### 3.6 Phenol isotope composition

In forest L$^-$ soils, the $\delta^{13}$C values were significantly higher in all vanillyl phenols ($p < 0.01$), as well as in the syringyl phenols
syringaldehyde ($p = 0.002$) and syringic acid ($p = 0.04$), compared to pasture L$^-$ soils (Fig. 4). Throughout the entire incubation
period, we observed a decrease of $\delta^{13}$C values in forest L$^-$ soils for vanillic acid and acetosyringone ($p = 0.04$, $p = 0.04$). In
pasture L$^-$ soils, $\delta^{13}$C decreased in vanillin ($p < 0.001$) and, surprisingly, showed a slight increase in piceol ($p = 0.01$). No
temperature trend was visible for any of the phenols. Litter addition led to a significant increase of $^{13}$C in both forest and
pasture soil for all phenols (all $p < 0.001$).

Over the entire incubation period, atomic % excess (APE) of $^{13}$C decreased for all phenols in forest L$^+$ soils, except for vanillic
acid, which showed no decline (Fig. 5). In pasture L$^+$ soils, we could detect a significant decrease in APE from start to end for
all phenols except for piceol ($p = 0.29$) and vanillic acid ($p = 0.78$). In forest L$^+$ soils, a stronger decrease in APE with higher
temperature was observed for vanillin (T$_{12.5}$ - T$_{20.5}$, $p = 0.01$), syringaldehyde (T$_{12.5}$ - T$_{20.5}$, T$_{16.5}$ - T$_{20.5}$; $p < 0.001$, $p = 0.02$),

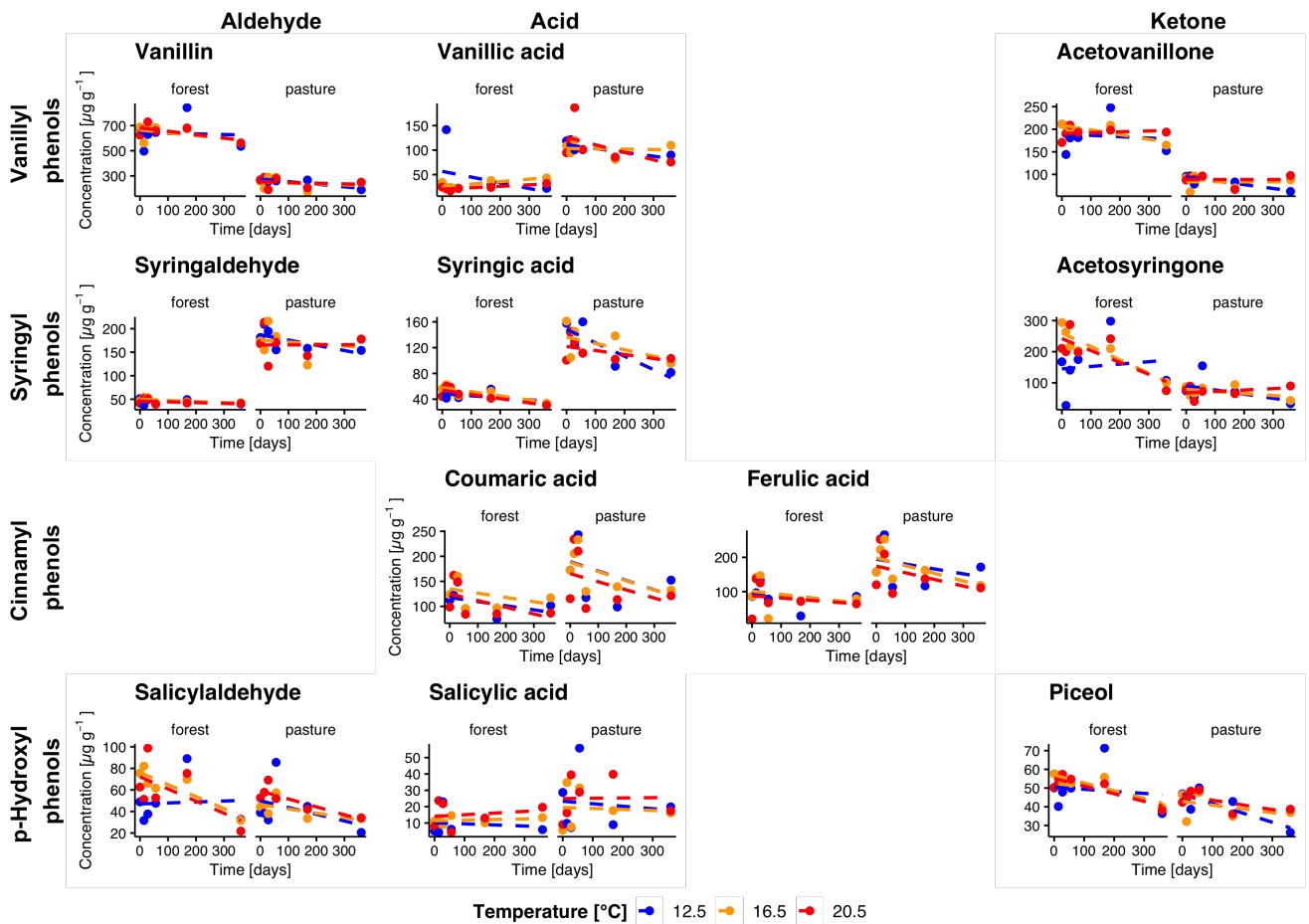

**Figure 2.** Temporal changes in individual phenol concentrations during incubation for forest and pasture soil samples without litter addition (L$^-$). Phenols are grouped horizontally into vanillyl, syringyl, cinnamyl, and p-hydroxyl compounds and vertically by functional class as aldehydes, acids, and ketones. Each point corresponds to the mean of the extracted samples (n = min. 3).

salicylic acid (T$_{12.5}$ - T$_{16.5}$, T$_{12.5}$ - $_{T20.5}$; $p = 0.02$, $p = 0.003$), coumaric acid (all temperatures, $p = 0.005$, $p < 0.001$, $p = 0.04$) and ferulic acid (T$_{12.5}$ - T$_{20.5}$, $p = 0.005$). In pasture L$^+$ soils, similar phenols showed a temperature dependency: Vanillin (T$_{12.5}$ - T$_{16.5}$, T$_{12.5}$ - T$_{20.5}$; $p = 0.03$, $p = 0.004$), syringaldehyde (T$_{12.5}$ - T$_{16.5}$, T$_{12.5}$ - T$_{20.5}$; $p = 0.003$, $p < 0.001$), coumaric acid (T$_{12.5}$ - T$_{16.5}$, T$_{12.5}$ - T$_{20.5}$; $p < 0.001$, $p < 0.001$).

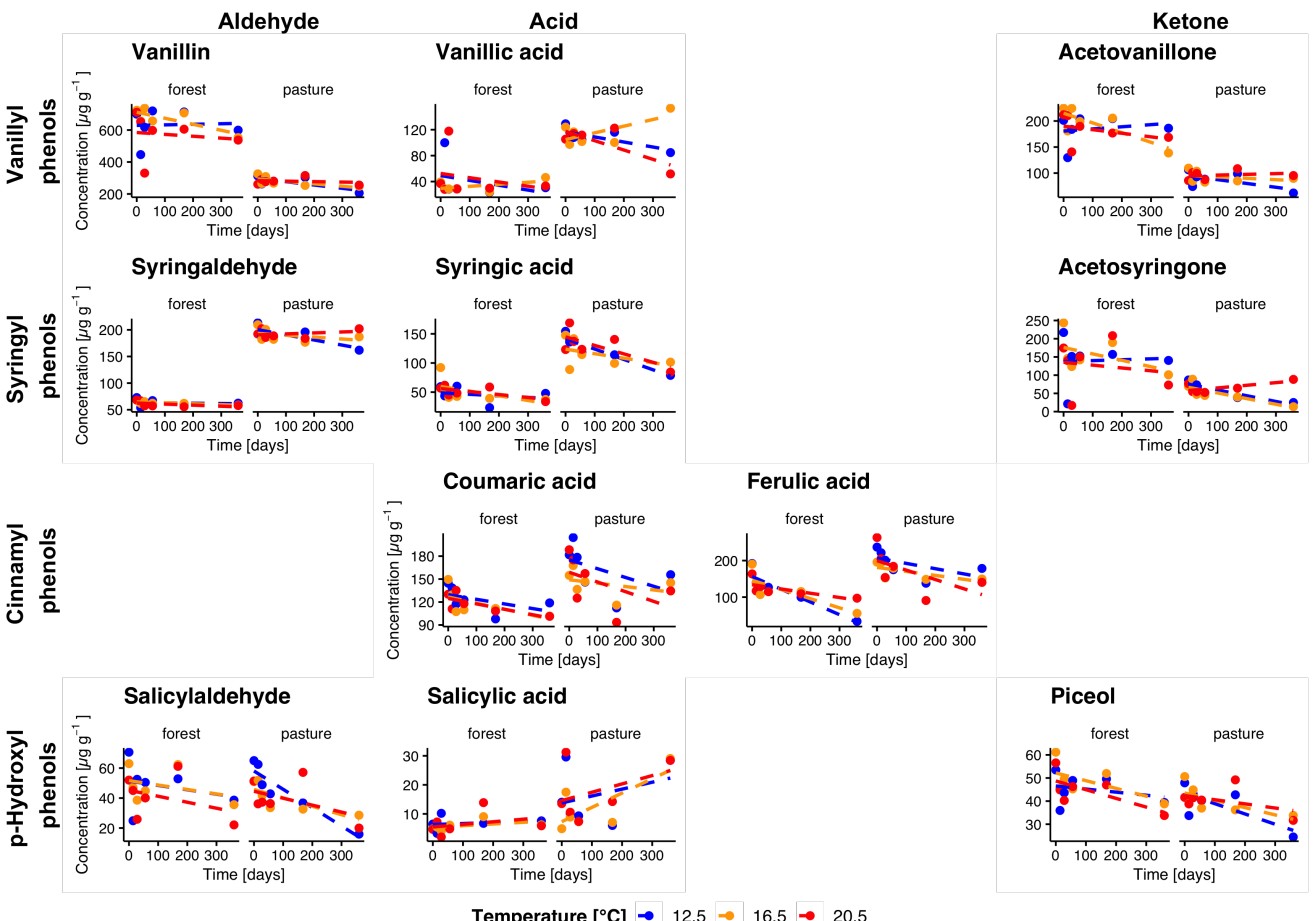

**Figure 3.** Temporal changes in individual phenol concentrations during incubation for forest and pasture soil samples with litter addition (L$^+$). Phenols are grouped horizontally into vanillyl, syringyl, cinnamyl, and p-hydroxyl compounds and vertically by functional class as aldehydes, acids, and ketones. Each point corresponds to the mean of the extracted samples for each treatment group (n = 4 for sampling at 14 days, n = 3 for other samplings).

## 4  Discussion

### 4.1  Decomposition of soil organic matter and litter at control temperature

The forest and pasture soil samples incubated at 12.5 °C provide a baseline to compare the decomposition of organic matter (OM) under current alpine conditions. These conditions represent the mean growing season temperature at the study site, a subalpine site in the Swiss Alps (Hiltbrunner et al., 2013; Speckert et al., 2023), and therefore enabling the assessment of the inherent differences between forest and pasture soils under controlled conditions.

Forest and pasture soils differed consistently in their carbon and nitrogen dynamics. The pasture soil exhibited slightly higher

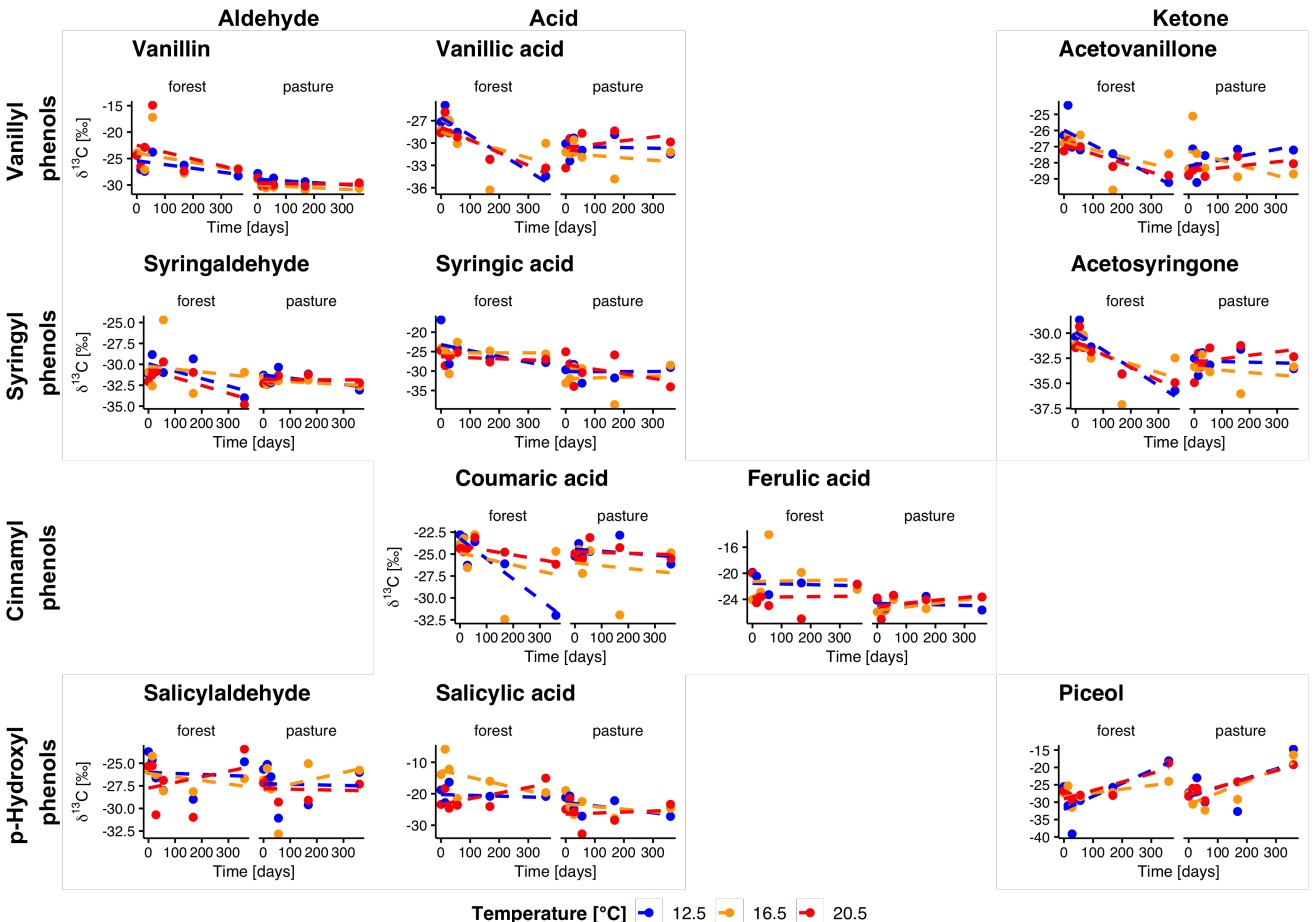

**Figure 4.** Evolution of natural abundance $\delta\,^{13}C$ in individual phenols over the incubation period for forest and pasture soil without litter addition (L⁻). Phenols are grouped horizontally into vanillyl, syringyl, cinnamyl, and p-hydroxyl compounds and vertically by functional class as aldehydes, acids, and ketones. Each point corresponds to the mean of the extracted samples (n = min. 3).

initial total organic carbon (TOC) concentrations compared to the forest soil, consistent with previous studies at the site (Hiltbrunner et al., 2013; Speckert et al., 2023) and other alpine soils in Switzerland (Hoffmann et al., 2014). The higher TOC in pasture soils can be explained by the type of OM inputs typical for this land use: Pasture soils receive regular inputs of above- and belowground biomass as well as manure deposition from grazing animals (Don et al., 2007; Conant et al., 2011; Speckert et al., 2023). These inputs are rich in labile and easily decomposable compounds such as sugars and proteins, leading to high TOC concentration (Rumpel, 2011). In contrast, forest soils primarily receive litterfall composed of needles and woody debris, which are rich in lignin and cellulose (Prescott and Vesterdal, 2021). These inputs accumulate in the litter layer and organic horizons (Speckert et al., 2023) and decompose more slowly than grass-derived OM because of their complex chemical structure (Prescott, 2010). Consequently, forest soils have a higher carbon and a lower nitrogen concentration com-

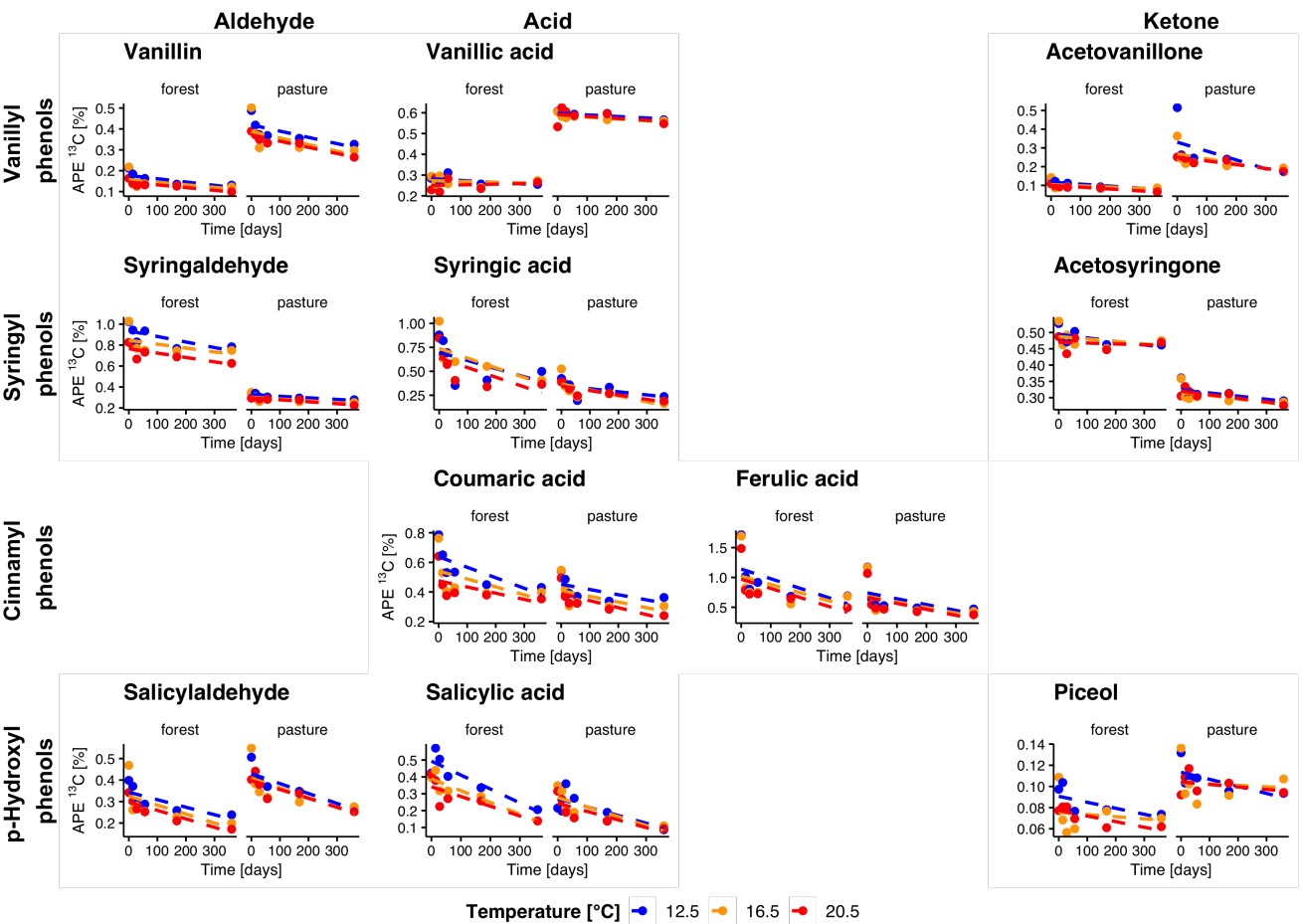

**Figure 5.** Evolution of natural abundance $\delta^{13}$C in individual phenols over the incubation period for forest and pasture soil with litter addition (L⁺). Phenols are grouped horizontally into vanillyl, syringyl, cinnamyl, and p-hydroxyl compounds and vertically by functional class as aldehydes, acids, and ketones. Each point corresponds to the mean of the extracted samples for each treatment group (n = 4 for sampling at 14 days, n = 3 for other samplings).

pared to pasture soils, leading to wider C/N ratios. This can constrain microbial decomposition by limiting nitrogen needed for growth and enzyme production (Melillo et al., 1982). Despite the higher initial TOC in the pasture soil, we observed a more pronounced decrease of TOC in the forest soil during incubation, which was almost double the decline observed in the pasture
soil. This suggests that the carbon lost in forest soils originated from more complex polymeric substances, e.g. lignin. At the same time, phenol concentrations also confirm stronger lignin losses: A significant decrease in lignin-derived phenols such as vanillyl and syringyl monomers was detected in forest soils, indicating active lignin decomposition (Hall et al., 2020). In contrast, decomposition of phenols was weaker in pasture soils without litter input, consistent with the assumed adaptation of the pasture's microbial community to fast-cycling labile carbon inputs (Breidenbach et al., 2022). The isotope data also revealed

differences in pasture soil: A slight decrease in $\delta^{13}$C over time suggests isotope fractionation during microbial decomposition, with preferential use of $^{13}$C-enriched compounds leaving the remaining SOM more depleted in $^{13}$C (Balesdent et al., 1987; Dijkstra et al., 2006). Similar modest decreases in bulk $\delta^{13}$C have been reported in other studies of grassland soils, where labile and $^{13}$C-heavier substrates are lost faster than more complex, isotopically lighter C pools such as lignin (Volk et al., 2018; Breidenbach et al., 2022). In summary, without fresh inputs, forest soils lost a larger fraction of stored carbon from lignin-rich

SOM, whereas decomposition in pasture soils slowed markedly once labile carbon became limited.

The addition of *Lolium perenne* litter at 12.5 °C shifted these dynamics in both soils. As expected, TOC and phenol concentrations initially increased in both pasture and forest soils due to the input of fresh litter. The labile fractions of the added grass material (Zhang et al., 2020) provided a readily available energy source, which stimulated microbial activity and enhanced decomposition not only of the added litter but also of more complex SOM components such as lignin (Kögel-Knabner, 2002).

This resulted in a stronger decline of TOC during incubation compared to the controls. In pasture soils, the decrease in TOC was more pronounced, indicating that the microbial community was likely well adapted to rapidly decompose grass litter. In forest soil compared to pasture soil, TOC decreased less strongly, but phenolic compounds declined more, pointing to a more intense decomposition of lignin-derived SOM. This pattern was also evident in the individual phenol groups: Cinnamyl monomers, the most easily decomposable group (Baumann et al., 2013), were decomposed rapidly in both soils, syringyl com-

pounds decreased more strongly in the forest soil. Vanillyl phenols, which are chemically more stable (Thevenot et al., 2010; Baumann et al., 2013), were decomposed only moderately, though still more in the forest than in the pasture soil. Together, these findings suggest that the pasture soil responded with faster decomposition of fresh inputs, while in the forest soil complex OM was more decomposable.

Total nitrogen (TN) increased after litter addition, more strongly in forest soils due to their lower initial TN and the higher

TN content of the added litter, but remained constant during the incubation period in both soils. This suggests that nitrogen was retained within microbial biomass or cycled within the system rather than lost (Schimel and Bennett, 2004; Mooshammer et al., 2014). The C/N ratio increased immediately after litter addition in both soils, but then decreased significantly during incubation, more strongly in pasture than in forest soil. This suggests that carbon was lost at a faster rate than nitrogen, consistent with microbial decomposition of the added litter and potentially also native SOM (Manzoni et al., 2008).

## 4.2 Influence of increasing temperature on the decomposition of organic matter

In soils without litter addition, temperature did not significantly influence the rate of SOM decomposition based on TOC concentrations in either soil type. It is surprising that no temperature trend was observed in TOC for these soils, given the general expectation that temperature would influence SOM decomposition (Nottingham et al., 2020; Soong et al., 2021). Carbon isotope composition also remained constant, with no observable temperature trend in either forest or pasture soil. Similarly, de-

composition rates of the different phenol groups showed no significant variation across temperature treatments. This highlights that without fresh, labile C, microbial communities were constrained in their ability to respond to warming (Allison et al., 2010; Fissore et al., 2013; Eberwein et al., 2015; Bradford et al., 2017).

In contrast, soils with litter addition exhibited a clear temperature-dependent decrease in TOC. This decrease has been seen in

many long-term warming experiments, some even despite an increased plant-derived organic matter input (San Román et al., 2024). Although total phenol concentrations in litter-amended soils did not differ significantly between temperature treatments, certain phenol groups showed increased decomposition with rising temperatures, both in concentrations and isotope signals. This effect is consistent with the increased decomposition of lignin phenols at higher temperatures, a process well-documented, where microbial activity and the temperature sensitivity of lignin-degrading enzymes enhance decomposition under elevated temperatures (Davidson and Janssens, 2006; Craine et al., 2010; Conant et al., 2011; D'Alò et al., 2021). Long-term warming experiments have also observed this increased decomposition of more complex organic matter, e.g. Tao et al. (2020); vandenEnden et al. (2021) and Zosso et al. (2023). Thus, the presence of fresh organic inputs not only stimulated decomposition but also amplified the apparent effect of temperature.

When looking at the temporal dynamics of decomposition, two distinct phases emerge. In the initial phase of decomposition, spanning the first 28 days, there is a rapid decline in TOC, phenol concentrations, and changes in $\delta^{13}$C across all treatments (see Supplement Table S1). This rapid phase is driven by the swift microbial utilization of readily available, labile C sources, such as simple sugars provided by the fresh litter addition in soils with litter addition or residual plant inputs in the soils without litter addition (Prescott and Vesterdal, 2021). Elevated temperatures amplify these processes by enhancing microbial metabolic rates and enzyme activities, leading to accelerated decomposition of these easily degradable substrates (Davidson and Janssens, 2006).

In the later phase of the incubation experiment, the rate of C loss diminishes significantly. Microorganisms in the soils deplete the easily accessible substrates and likely shift to metabolizing more complex compounds like lignin-derived phenols, which are inherently more resistant to microbial breakdown (Kögel-Knabner, 2002). This transition results in a slower overall decomposition rate and reduced temperature sensitivity, as the decomposition of complex organic molecules depends more on specialized enzymes and microbial community adaptations than on temperature alone (Feng and Simpson, 2008; Thevenot et al., 2010; Conant et al., 2011; Manzoni, 2017). Moreover, the physical and chemical protection of more complex SOM within soil aggregates further shields from microbial access (Schmidt et al., 2011; Lehmann and Kleber, 2015). This protective mechanism reduces the sensitivity of stabilized SOM to temperature increases (Qin et al., 2019).

The temperature trend is visible for more phenols in the forest soil samples than in the pasture soil. This is likely another indicator for the initial presence of different soil microbial communities adapted to different input of plant-derived OM. An analysis of the microbial communities at our site observed increased concentrations in Gram[+] bacteria and actinobacteria and a higher fungi-to-bacteria ratio in the forest soil compared to the pasture soil (Hiltbrunner et al., 2012; Speckert et al., 2025). This forest soil community may sustain decomposition more effectively over extended periods, even when labile substrates are scarce. In contrast, pasture soils, which are typically adapted to more frequent inputs of labile OM such as root exudates and grass litter, responded strongly to warming during the early phase but experienced a sharper decline in microbial activity once these substrates were exhausted. If their microbial communities are less capable of decomposing more complex compounds, this can lead to a more significant slowdown in long-term decomposition (Fontaine et al., 2007; Vanhala et al., 2008).

These differences between forest and pasture soils illustrate that warming effects were most pronounced during the initial, labile-C driven phase of decomposition during the first month of incubation, but became much weaker once these easily

degradable substrates were depleted. This shift from an early to a later decomposition phase is consistent across our treatments and underlines the importance of substrate availability in shaping decomposition responses. In summary, our results show that the apparent temperature sensitivity of SOM decomposition is strongly phase-dependent: warming enhances rapid early-phase decomposition of labile C, but its effect diminishes during late-phase decomposition when more complex substrates dominate. This demonstrates that substrate availability, rather than temperature alone, controls long-term SOM decomposition dynamics in subalpine forest and pasture soils.

### 4.3 Decomposition of SOM in future alpine ecosystems

Our results provide indirect evidence that the two alpine pasture and forest soils host functionally different microbial communities. This has also been shown in other studies, with more bacteria-dominated alpine grassland soils and forest soils with a higher abundance of fungal microorganisms (Djukic et al., 2010). These different communities react differently to the influence of rising temperatures and changes in the input of litter. Vegetation changes such as afforestation or shrub encroachment, as they occur in many alpine areas, can therefore have a strong influence on the carbon cycle. Additionally, the projected strong temperature increase in alpine region in the next decades will have an especially large influence on soil carbon cycling. Our findings implicate a likely increase in decomposition of SOM in alpine regions. The question arises as to how these temperatures affect the growth of the vegetation and thus the plant derived OM input in these soils. With an earlier onset of snow melt due to climate change (Rogora et al., 2018), the time window in which SOM can be decomposed on a large scale is also increasing (Magnani et al., 2017). Alpine coniferous forests have the largest amount of litter fall in late summer and autumn (Pausas, 1997). A large proportion of the easily degradable carbon contained therein is presumably decomposed within a few weeks, as our results suggest: We observed a phase of strong C loss within the first month of incubation across treatments, which would correspond to the fast decomposition of simple substrates soon after litterfall. In addition, an increase in temperature seems to lead to earlier senescence of litter and thus a longer decomposition phase before the onset of snow cover (Ernakovich et al., 2014; Möhl et al., 2022). At the beginning of spring with the onset of snow melt, the SOM therefore consists largely of rather complex compounds. If we consider this from the perspective of our results, in which we observed a still considerable decomposition of SOM in the forest soil without fresh litter, an increased decomposition of SOM in the course of a year can be expected. With additional input of fresh litter, the increased temperatures will also lead to a further increase in the decomposition of SOM, not only of simple compounds but also of more complex polymers such as lignin. Even though a one-year laboratory experiment allows only limited conclusions, our results seem to support the hypothesis of increased SOM decomposition in alpine regions with increasing temperatures, as has already been found in many other studies in alpine regions for both grassland (Chen et al., 2024) as well as forest ecosystems (Albrich et al., 2023). NPP generally increases with rising temperatures (Rustad et al., 2001), especially in alpine regions (Wang et al., 2023), which could lead to an increased input of litter into the soil. However, this would also increase the availability of fresh organic material and thus stimulate the decomposition, as we have seen in our experiment. Therefore, while many alpine soils are still sinks of C, they could therefore potentially develop into sources of C into the atmosphere as temperature rises.

## 5   Conclusions

Our study emphasises that substrate quality, in particular the availability of labile carbon, is the most important driver of SOM decomposition in the examined alpine forest and pasture soils, outweighing the influence of temperature alone. While temperature increases decomposition, this effect is only enhanced in the presence of labile carbon, especially originating from fresh litter. Contrary to our initial hypothesis, the alpine forest soil (with its lignin-rich material) showed higher carbon loss without litter presumably due to specialised microbial communities, while the pasture soil without labile carbon input exhibited only limited decomposition. The addition of grass litter greatly accelerated decomposition in both investigated soils, confirming our hypothesis that fresh input acts as a priming agent. Decomposition occurs in the short term due to the decomposition of more labile carbon, but slows down in the long term as complex substrates such as lignin dominate. Our findings imply that if alpine ecosystems experience warming along with shifts in vegetation that increase labile litter inputs, SOM decomposition could accelerate substantially. This scenario could potentially reduce the soil C sink strength of similar high elevation soils and in the worst case turning a C sink into a source. Accounting for carbon availability in climate modelling is essential for a better prediction of soil carbon dynamics under future climate scenarios.

*Author contributions.*  DP: Conceptualization, Data Curation, Formal Analysis, Investigation, Visualization, Writing – Original Draft. TCS: Conceptualization, Investigation, Writing – Review & Editing. YAB: Methodology, Resources, Writing – Review & Editing. GLBW: Conceptualization, Funding Acquisition, Investigation, Methodology, Project Administration, Resources, Supervision, Writing – Review & Editing.

*Competing interests.*  The authors declare that they have no conflict of interest.

*Acknowledgements.*  We thank Carrie L. Thomas for her invaluable assistance during the sampling campaign and throughout the incubation experiment. We also extend our gratitude to Barbara Siegfried, Sonja Eisenring, and Nadja Hertel for their support during the laboratory work. We thank Kate Buckeridge and two anonymous reviewers for their constructive comments and suggestions, which greatly improved the quality of this manuscript. This research was funded by the Swiss National Science Foundation (SNSF) under grant no. 188684, as part of the IQ-SASS project (Improved Quantitative Source Assessment of Organic Matter in Soils and Sediments using Molecular Markers and Inverse Modelling).

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
