# Peer review of "Availability of labile carbon controls the temperature-dependent response of soil organic matter decomposition in alpine soils"

_EGUsphere, 2025_

## Author Response (AR1)

**Response to Reviewers**

We thank both reviewers and the editor for their thoughtful and constructive feedback on our manuscript. We greatly appreciate the positive comments regarding the design, relevance, and novelty of our study. Following the reviewers' suggestions, we revised the manuscript to improve clarity, conciseness, and scientific precision. Below, we provide a point-by-point response to each comment and indicate the corresponding changes in the revised manuscript.

**Response to the comments of Anonymous Referee #1**

We thank the reviewer for the thoughtful and constructive feedback on our manuscript. We appreciate the positive comments regarding the design, relevance and the novelty of our study. In response to the specific suggestions, we revised the manuscript to improve clarity, conciseness and scientific precision. Below, we address each comment and outline the changes in the manuscript.

1) The discussion is long and would benefit from combining a couple related sections with more concise language.
We are grateful for this suggestion; we agree that the discussion is quite lengthy. We combined some of the chapters to minimise repetition, especially chapters 4.1 and 4.2 as well as 4.3 and 4.4 have been linked and streamlined, as here the discussion was partially repetitive. The merge of the chapters will lead to easier reading.

2) The language in the discussion section about microbial communities differing should be revised to reflect the speculative nature of those comments, since the microbes themselves were not directly measured. For example, in L411, perhaps state "our results provide indirect evidence..."
We agree that statements about microbial communities should be framed more cautiously. We revised the relevant sentences throughout the discussion to clearly reflect the indirect nature of our evidence.

3) The hypotheses should be reworded for clarity. One uses the word "potentially" twice, distracting from the main point. A hypothesis should be more direct.
Thanks a lot for this suggestion, we reworded the hypotheses, accordingly. They are now more direct and thus hopefully clearer.

4) Are there any suggestions for future research needs, such as enzyme kinetics? Perhaps an explanation for the lack of temperature response without added litter? Have other studies examined microbial dynamics in these specific soils?

Enzyme kinetics might indeed be an interesting thing to look at, as they are often highly temperature sensitive, but the response to litter (labile C) availability is still unclear. So far, we did not investigate this, but we also conducted additional analysis of the incubated soils. We measured the concentration and C isotope composition of phospholipid fatty acids (PLFA) to gain insights into changes of microbial communities and the incorporation of the litter into the different microbial groups. This data will be included in another manuscript that is currently in preparation. We therefore did not explicitly add this to the revised manuscript.

5) L14: reservoirs not reservoir

Thank you, this has been changed as suggested.

6) L28 "are discussed to be vulnerable" is awkward phrasing

Thank you, we agree and have changed the phrasing in the revised hypotheses to "[…]are especially vulnerable to warming".

We hope these revisions address the raised concerns and we are sure that the changes have improved the overall quality and readability of the manuscript. We thank the reviewer again for the valuable insights.

**Response to the comments of Anonymous Referee #2**

We thank the reviewer for the thoughtful and constructive comments on our manuscript. We are pleased that they found the study to be well-designed, well-written, and relevant to pressing questions about soil C dynamics under warming. Below, we respond to each point in detail and outline the revisions made to improve the manuscript.

1) The authors may wish to consider shortening some sections, as the manuscript as a whole is quite long.
We are grateful for this comment. We agree, especially the discussion seems to be lengthy. We combined discussion chapters 4.1. & 4.2 as well as 4.3 and 4.4 to reduce redundancy, emphasise important points more and thus also shorten the manuscript and improve the readability.

2) L4 Abstract – why 'even'?
We agree, this doesn't make sense, and we have removed the 'even' in the revised abstract.

3) The abstract is too general and should present more of the study's results.
We revised the abstract to include more specific results. We summarized the key quantitative findings, including temperature effects on TOC and differences in decomposition between soils with and without litter. The revised version now highlights important findings now with the most important results.

4) L7-9 'Without fresh litter, SOM decomposition was limited, suggesting that substrate availability in combination with temperature increase plays a greater role in microbial activity than temperature alone." This is obvious. Please be more precise and explain how big these effects were.
We agree and have revised this sentence in the process of rewriting the abstract with more of the study's. Now the size of the effect is also mentioned.

5) First paragraph of Introduction needs for more up-to-date references
We appreciate this suggestion and have updated the introduction with several recent references on SOM decomposition and warming impacts in alpine regions, e.g. Chen et al. (2024) Bright et al. (2025), Bonfanti et al. (2025). This improves the contextual framing with newest scientific evidence and supports the relevance of our study.

6) L91 M&M 30 kg? This means that the sampling sites were quite close to the roads to enable the transportation of such a large sample. It may affect sampling sites (eutrophication, compaction etc.). Please discuss this issue
We appreciate this critical observation. The, indeed, large samples were transported in several batches from soil pits to the small alpine path (no paved road), where samples were collected in a four-wheel drive vehicle. To minimize the effect of the road, we either sampled in a large distance above it (pasture) or at a site that was not influenced by the road (forest). We therefore think that any effect of the road was minimized by the careful site selection and the homogenization of the samples. However, we guess that such information is redundant in the M&M section and have excluded this, unless the reviewer or the editor see this as a crucial point that is worth it mentioning.

7) Table A1 – I'm not sure if this is necessary – the table is huge! However, I cannot find the Guide for Authors, so perhaps it is required or accepted by the publisher.
We thank the reviewer for raising this concern. Table A1 provides an overview of the incubation experiment parameters and the measured soil properties across all treatments and time points. This data supports key interpretations regarding litter-derived C incorporation and SOM decomposition dynamics. We believe this level of detail is important for transparency and reproducibility. However, for the final manuscript, we have moved the Table A1 to the Supplementary Material where it is now Table S1.

We thank the reviewer again for their helpful feedback, which will strengthen the manuscript both scientifically and structurally. We believe the revisions improved the manuscript's precision, clarity and conciseness, and we hope it will meet the reviewer's expectation for publication in SOIL.

References

Bright, K., Dienes, B., Keiluweit, M., Rixen, C. & Aeppli, M. *Climate change impacts on organic carbon cycling in European alpine soils.* Soil Biology and Biochemistry 210, 109891 (2025), https://doi.org/10.1016/j.soilbio.2025.109891

Bonfanti, N., Clément, J., Münkemüller, T., Barré, P., Baudin, F., Poulenard, J. *Prolonged warming leads to carbon depletion and increases nutrient availability in alpine soils.* Applied Soil Ecology 213, 106239 (2025). https://doi.org/10.1016/j.apsoil.2025.106239

Chen, C., Wang, L., Xia, W., Qiu, K., Guo, C., Gan, Z., Zhou, J., Sun, Y., Liu, D. Li, W. & Wang, T. *Molecular interaction induced dual fibrils towards organic solar cells with certified efficiency over 20%.* Nature Communications 15, 6865 (2024). https://doi.org/10.1038/s41467-024-51359-w

**List of Changes**

**Major changes**

Abstract:

- L1 – 12: Changed the abstract. It now less general, direct results have been added and the conclusion part of it was reworded to increase clarity and flow.

Introduction

- L61 – 74: The hypotheses were reworded to improve clarity. They are now more direct without overusing "potentially"

Discussion

- L258 – 304: Former subchapters 4.1 and 4.2 discussing the decomposition of SOM and litter, respectively, have been merged into one subchapter that discusses the decomposition of both. This helps to reduce repetition, as many explanations for the observed decomposition patterns in both SOM and litter can be explained by similar processes.

- L305 – 354: Former subchapters 4.3 and 4.2 discussing the influence of increasing temperature and the temporal component of the decomposition have been streamlined and merged. This reduced redundancy, linked different findings better and shortened the discussion to increase readability. Additionally, statements on the influence of different microbial communities have been softened to reflect the speculative, indirect nature of our interpretations and to distinguish between our measured results and broader ecological inferences.

Appendix

- Figures A1 and A2 as well as Tables A1 – A7 have been moved from Appendix to Supplement.

Minor changes

- L14: changed "reservoir" to "reservoirs"
- L22: added the reference "Chen et al., 2024"
- L27: added the reference "Bright et al., 2025"
- L29: Removed "Particularly"
- L29: changed "discussed to be" to "especially"
- L29f: added the reference "Bonfanti et al., 2025"
- L88: replaced minus by hyphen
- L90: added dash between number and unit
- L97: replaced minus by hyphen
- L103f: replaced minus by hyphen
- L107: changed figure reference to the supplement
- L113: changed figure reference to the supplement
- L124: removed capital A
- L188: changed figure reference to the supplement
- L193: changed figure reference to the supplement
- L196: changed figure reference to the supplement
- L197: changed figure reference to the supplement
- L203: changed figure reference to the supplement
- L209: replaced minus by hyphen x2

- L211f: changed figure reference to the supplement
- L217: changed figure reference to the supplement
- L225: changed figure reference to the supplement
- L356: added "indirect"

---

## Author Response (AR2)

**Response to the comments of Anonymous Referee #1**

We would like to thank the reviewer once again for their time and for providing renewed feedback. We greatly appreciate the suggested changes and have implemented them as requested. We hope that the revised manuscript now meets the standards for publication in SOIL. The specific changes are detailed below.

1) L15-16: To me, it's redundant to include the negative before the number since you're saying "losses of", but others may have different opinions.

Thank you for this comment, we removed the negative on these lines.

2) L17: awkward wording "decomposition of harder decomposable SOM"

We changed the wording to "decomposition of more complex SOM".

3) L64: remove "thus as follows"

Removed as requested.

4) L277: reword to "enabling the assessment of"

We reworded this accordingly.

5) L304: reword to "phenol concentrations also"

We changed the word order as requested.

6) L305: "indicating" instead of "pointing to"

Changed as requested.

7) L355: "declined more" instead of "declined stronger"

Changed the wording as requested.